# High throughput detection and genetic epidemiology of SARS-CoV-2 using COVIDSeq next-generation sequencing

Rahul C. Bhoyar[1], Abhinav Jain[1,2], Paras Sehgal[1,2], Mohit Kumar Divakar[1,2], Disha Sharma[1], Mohamed Imran[1,2], Bani Jolly[1,2], Gyan Ranjan[1,2], Mercy Rophina[1,2], Sumit Sharma[1], Sanjay Siwach[1], Kavita Pandhare[1], Swayamprabha Sahoo[3], Maheswata Sahoo[3], Ananya Nayak[3], Jatindra Nath Mohanty[3], Jayashankar Das[3], Sudhir Bhandari[4], Sandeep K. Mathur[4], Anshul Kumar[4], Rahul Sahlot[4], Pallavali Rojarani[5], Juturu Vijaya Lakshmi[5], Avileli Surekha[5], Pulala Chandra Sekhar[5], Shelly Mahajan[6], Shet Masih[6], Pawan Singh[6], Vipin Kumar[6], Blessy Jose[6], Vidur Mahajan[6], Vivek Gupta[7], Rakesh Gupta[7], Prabhakar Arumugam[1,2], Anjali Singh[1,2], Ananya Nandy[1,2], Ragavendran P. V.[1,2], Rakesh Mohan Jha[1,2], Anupama Kumari[1,2], Sheetal Gandotra[1,2], Vivek Rao[1,2], Mohammed Faruq[1,2], Sanjeev Kumar[1,2], Betsy Reshma G.[1,2], Narendra Varma G.[1], Shuvra Shekhar Roy[1,2], Antara Sengupta[1,2], Sabyasachi Chattopadhyay[1,2], Khushboo Singhal[1,2], Shalini Pradhan[1], Diksha Jha[1,2], Salwa Naushin[1,2], Saruchi Wadhwa[1,2], Nishu Tyagi[1,2], Mukta Poojary[1,2], Vinod Scaria[1,2]*, Sridhar Sivasubbu[1,2]*

1 CSIR Institute of Genomics and Integrative Biology (CSIR-IGIB), New Delhi, India, 2 Academy for Scientific and Innovative Research, Human Resource Development Centre Campus, Ghaziabad, Uttar Pradesh, India, 3 Institute of Medical Sciences and SUM Hospital, Siksha "O" Anusandhan (Deemed to be University), Bhubaneswar, Odisha, India, 4 Sawai Man Singh Medical College, Jaipur, Rajasthan, India, 5 Kurnool Medical College, Kurnool, Andhra Pradesh, India, 6 Center for Advanced Research in Imaging, Neuroscience & Genomics, New Delhi, Delhi, India, 7 Government Institute of Medical Sciences, NOIDA, Uttar Pradesh, India

* s.sivasubbu@igib.res.in (SS); vinods@igib.res.in (VS)

**Data Availability Statement:** Raw datasets are available at NCBI short Read Archive with Project ID PRJNA655577.

## Abstract

The rapid emergence of coronavirus disease 2019 (COVID-19) as a global pandemic affecting millions of individuals globally has necessitated sensitive and high-throughput approaches for the diagnosis, surveillance, and determining the genetic epidemiology of SARS-CoV-2. In the present study, we used the COVIDSeq protocol, which involves multiplex-PCR, barcoding, and sequencing of samples for high-throughput detection and deciphering the genetic epidemiology of SARS-CoV-2. We used the approach on 752 clinical samples in duplicates, amounting to a total of 1536 samples which could be sequenced on a single S4 sequencing flow cell on NovaSeq 6000. Our analysis suggests a high concordance between technical duplicates and a high concordance of detection of SARS-CoV-2 between the COVIDSeq as well as RT-PCR approaches. An in-depth analysis revealed a total of six samples in which COVIDSeq detected SARS-CoV-2 in high confidence which were negative in RT-PCR. Additionally, the assay could detect SARS-CoV-2 in 21 samples and 16 samples which were classified inconclusive and pan-sarbeco positive respectively suggesting that COVIDSeq could be used as a confirmatory test. The sequencing approach also enabled insights into the evolution and genetic epidemiology of the SARS-CoV-2 samples. The samples were classified into a total of 3 clades. This study reports two lineages

**Funding:** Authors acknowledge funding for the work from the Council of Scientific and Industrial Research (CSIR), India through grants CODEST and MLP2005. The funders had no role in study design, data collection and analysis, decision to publish, or preparation of the manuscript.

**Competing interests:** The authors have declared that no competing interests exist.

B.1.112 and B.1.99 for the first time in India. This study also revealed 1,143 unique single nucleotide variants and added a total of 73 novel variants identified for the first time. To the best of our knowledge, this is the first report of the COVIDSeq approach for detection and genetic epidemiology of SARS-CoV-2. Our analysis suggests that COVIDSeq could be a potential high sensitivity assay for the detection of SARS-CoV-2, with an additional advantage of enabling the genetic epidemiology of SARS-CoV-2.

## Introduction

Coronavirus Disease 2019 (COVID-19) has emerged as a global epidemic affecting millions of individuals globally and imposes a huge burden on the socio-economic welfare and healthcare systems of nations. At present, the need for assays for rapid detection for diagnosis and surveillance, understanding the genetic epidemiology and evolution of the virus would be central for managing the spread of the epidemic [1, 2]. The advantage of quick sequencing of the SARS-CoV-2 genome led to the development of polymerase chain reaction (PCR) based diagnostic assays that leveraged rapid identification of infected individuals to get fast medical support or quantization essential to both patient management and incidence tracking [3]. Identification of early imported cases in France helped to prevent immediate secondary transmission [4]. Singapore's enhanced surveillance and containment strategy also led to the suppressed expansion of SARS-CoV-2 [5]. On similar grounds, The Royal College of General Practitioners (RCGP) Research and Surveillance Centre (RSC) have rapidly expanded their national surveillance system to combat SARS-CoV-2 [6]. Coupled with a highly accurate and high-throughput method of detection, this approach is expected to become more effective in dealing with COVID-19.

A number of approaches have been widely used for the detection of SARS-CoV-2 from clinical samples. Some of these approaches have also been adapted to enable higher throughputs. These methods are majorly subdivided into antigen-antibody based serological assays, nucleic acid-based amplification assays, and sequencing-based assays. While serological assays are rapid detection tests, they have low sensitivity and specificity [7]. Nucleic acid-based amplification such as quantitative real-time PCR (qRT-PCR) has been the gold standard in detection and diagnosis, but a negative RT-PCR does not eliminate the possibility of infection in clinically suspected cases [8]. Such results should be carefully interpreted to avoid false-negative reporting [9]. Moreover, these tests have been developed for diagnostic purposes and do not provide much information on the nature of the virus, its genetic information, and evolutionary pattern. In this regard, recently developed next-generation sequencing-based methods are potentially a good alternative for the detection of SARS-CoV-2 [10].

The rapid advancement of next-generation sequencing technology and analysis methods has enabled understanding the genetic makeup of SARS-CoV-2 and interpreting its evolutionary epidemiology. Viral RNA sequencing from the initial cluster of cases deciphered the full genome sequence of SARS-CoV-2 [11]. This led to other sequencing-based studies for detailed genomic characterization of the virus [12]. Combined genetic and epidemiological studies have been suggested to provide insights into the spread of the infection, evolutionary patterns, and genetic diversity of the virus, [1, 2] for effective management and preventive measures. Genomic surveillance coupled with agent-based modeling in Australia has been observed as an excellent approach to investigate and regulate COVID-19 transmission [13]. Towards these efforts, several openly available databases have also been developed such as the Global

Initiative on Sharing All Influenza Data (GISAID) that facilitates rapid and open sharing of SARS-CoV-2 genome sequences [14]. Thus, along with detection, sequencing-based methods may also provide an added advantage of understanding the genetic epidemiology of the outbreak.

In the present study, we describe the application of the COVIDSeq protocol recently approved by the United States Food and Drug Administration (US FDA) for clinical use. This protocol envisages high-throughput detection and genetic epidemiology of SARS-CoV-2 isolates using a multiplex PCR amplicon-based enrichment followed by barcoding with a throughput of 1536 samples in a single sequencing run using NovaSeq S4 flow cell. Our analysis suggests that the COVIDSeq protocol could be a sensitive approach for detection with additional insights offered through genetic epidemiology with respect to the genetic lineages. To the best of our knowledge, this is the first real-life evaluation of COVIDSeq protocol.

## Results

The sample panel consisted of a total of 752 samples. Among these, 655 (87.1%) were SARS-CoV-2 positive on RT-PCR as per the diagnostic guidelines laid out by the Indian Council for Medical Research (ICMR). We included 19 samples that were RT-PCR negative (2.5%) and 43 samples (5.7%) were categorised as pan-sarbeco, since they were positive for the E gene primers only. E gene is conserved across all betacoronaviruses and its detection in the sample indicates the presence of either SARS-CoV or SARS-CoV-2 or other bat-CoVs. A total of 35 samples (4.6%) were considered inconclusive as the samples had one of the two genes (i.e. ORF1ab gene) tested positive. In these samples, the E gene was tested negative hence these samples were termed inconclusive. Apart from this, the sequencing panel consisted of 8 COVIDSeq Positive Control HT (CPC HT) and 8 no template control (NTC) as internal process controls making total samples to 768. The quality of the pooled library was checked by agarose gel electrophoresis and fragment analyzer which showed the fragment size to be around 300bp. The panel was sequenced in technical duplicates making it a total of 1536 samples in total. The sequencing was performed for 36 cycles. The runtime of the sequencer was 11 hours. Sequencing generated a total of 705.64 Gb of data with 86.90% cluster passing filter and 95.62% above the quality cutoff of (QC) 30. Sequencing generated on an average of approximately 8.4 million reads for the 1,536 samples. The schematic of the COVIDSeq pipeline is depicted in Fig 1.

The COVID-19 detection was performed using the DRAGEN COVIDSeq Test pipeline that implements SARS-CoV-2 detection criteria of at least 5 SARS-CoV-2 targets to be considered as positive. Out of the 1,504 samples, the DRAGEN COVIDSeq Test pipeline successfully annotated 1,352 samples. Further 136 samples were classified as undetected, and 16 failed the internal quality check. This corresponds to 676 unique samples in which SARS-CoV-2 was detected, 68 unique samples in which SARS-CoV2 was undetected, and 8 unique samples that failed the assay. There was no discordance in the annotations between any of the 752 sample duplicates considered, suggesting a cent percent concordance in the detection. The total runtime for the DRAGEN COVIDSeq pipeline was 374 minutes. The stepwise runtime is summarised in the S1 Table.

All samples were also further considered for in-depth alignment and on average 8.4 million raw reads were generated for 1,536 samples, which were trimmed at base quality Q30 and read length of 30 bps that lead to an average of 7.9 million reads. The trimmed reads were further aligned to the human reference genome (GRCh38/hg38) and SARS-CoV-2 genome (NC_045512.2). On average we found 2.4 million human reads with a mapping percentage of 30.73% and 5.04 million SARS-CoV-2 reads with a mapping percentages of 63.89%

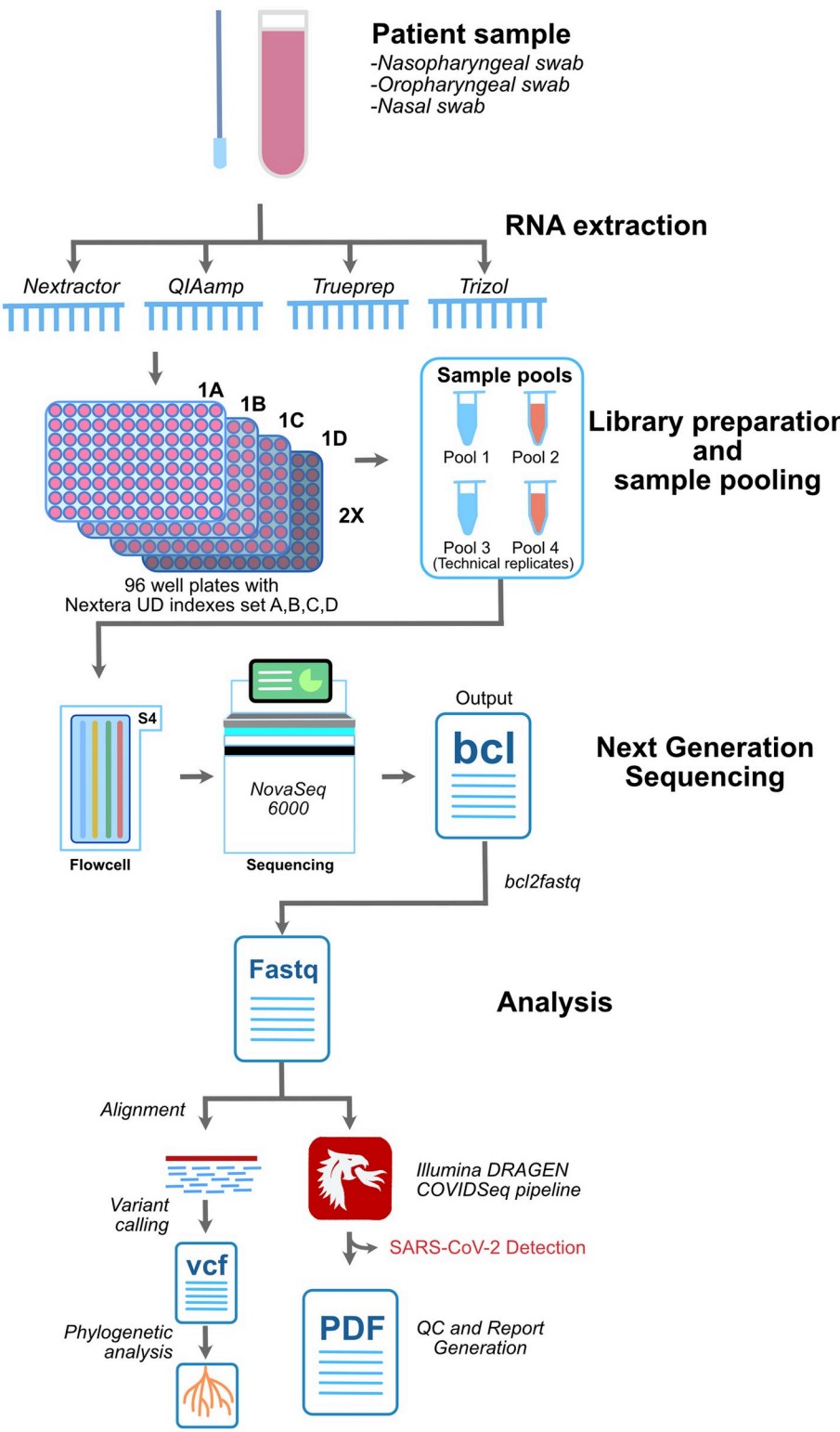

**Fig 1. Schematic summary of the analysis in this study.** The methodology adopted in the sampling, library preparation, sequencing and analysis involving custom based pipeline and COVIDSeq pipeline employed in this study.

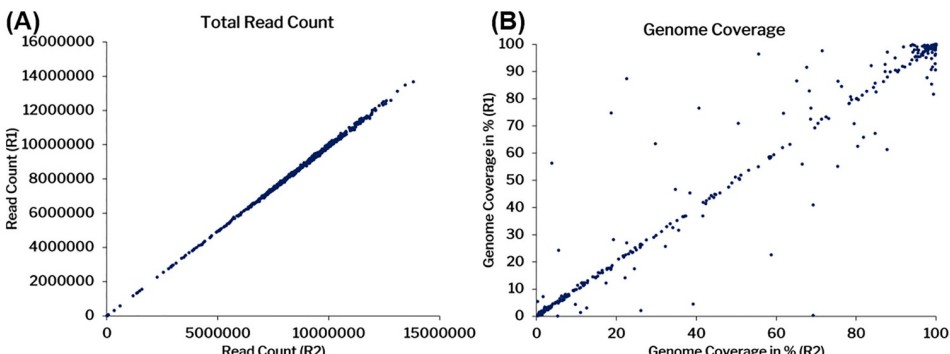

**Fig 2. Concordance among replicate samples considered in the analysis.** A) Total number of read counts B) The coverage percentage among replicates. R1-Replicate 1, and R2- Replicate 2.

respectively. The unmapped reads from the human aligned files were extracted and mapped to the SARS-CoV-2 reference genome (NC_045512.2) to increase its specificity and 4.4 million such reads (79.34%) mapped to it with 6322X coverage. Fig 2 summarises the concordance of aligning reads as well as genome coverage across the duplicates. The data has been summarized in the S2 Table.

The mean coverage was also plotted for all the samples across 98 PCR amplicons covering the whole SARS-CoV-2 genome represented in Fig 3. The mean coverage across the amplicons was ~14256x for the positive samples considered (706 samples with genome coverage >5%). We have found 20 amplicons had coverage ±2 standard deviations (SD) of this value, out of which 16 amplicons had coverage <2 SD and 4 amplicons had >2 SD.

The technical duplicates had a correlation coefficient of 0.99 (p-value < 0.00001) for reads and 0.984 (p-value < 0.00001) for the coverage.

For further genome assembly and variant calling, the alignment files were merged and variants were called using VarScan. Only 495 samples that had at least 99% of the genome covered were considered for the variant call. Out of these 495 samples, 91 samples were from Andhra Pradesh, 63 from Odisha, and 341 samples were from Delhi.

The analysis identified a total of 1,143 unique variants. 73 genetic variants were found to be novel in comparison with other Indian and global genome data and were reported for the first time. The median for the number of variants called was 12. The distribution of the variants per

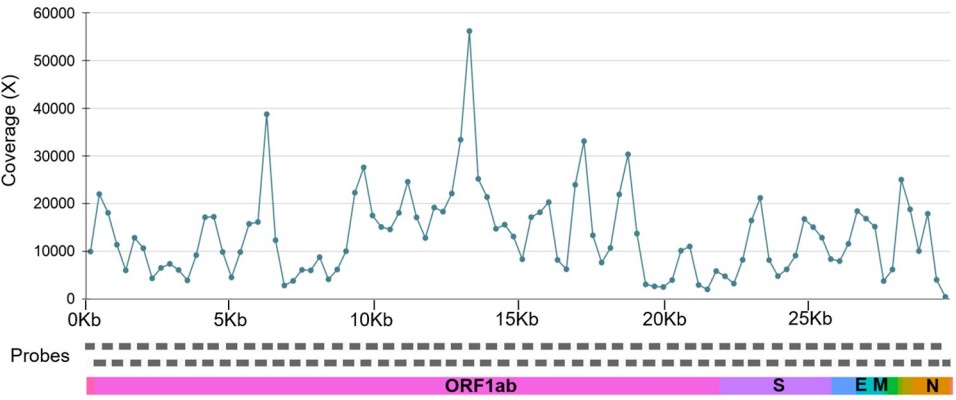

**Fig 3. The line plot for the mean coverage of SARS-CoV-2 genome.** The mean coverage for the 98 amplicons across the SARS-CoV-2 genome.

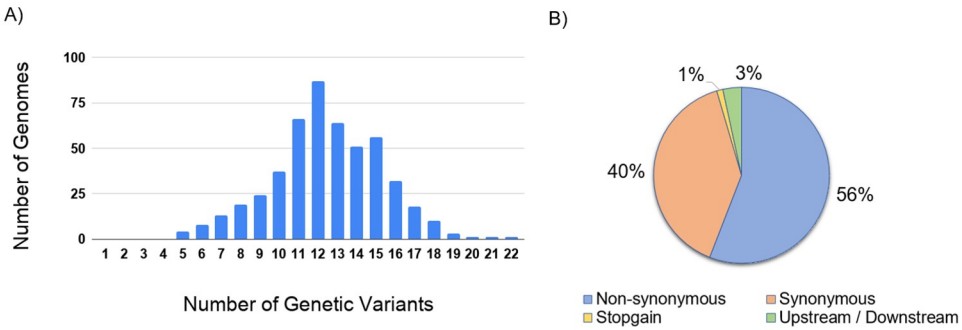

**Fig 4. Variant number per genome and their annotation.** (A) Distribution of variants in the genomes with ≥ 99% coverage (B) Summary of the variant annotations.

genome is summarised in Fig 4A. Of the 1,143 unique variants, a total of 1,104 variants were in the exonic region and 39 were in the downstream or upstream region. Of the 1,104 exonic variants, 639 variants were non-synonymous while 452 were synonymous. A total of 13 were found to be stopgain. The variant annotation data is summarized in Fig 4B. The detailed annotations of the variants have been tabulated in the S3 Table. On evaluating the functional importance of the variants, there are 38 genetic variants with potential functional implications reported in the literature. The variants functional annotation for 38 variants are listed in the S4 Table.

Analysis of frequency of the variants across the genomes revealed a total of 89 variants that had a frequency ≥ 1% and were polymorphic. The variants were also mapped across the genes. The ORF1ab gene had the largest number of variants. Normalised for the length of the genes, the ORF3a gene had the highest number of variants. Similarly, for non-synonymous variants, the ORF1ab gene had the highest number of variants and ORF3a had the highest normalized for the length of the gene.

To get an insight into the genetic epidemiology, the genomes were analyzed for their phylogenetic distribution. Phylogenetic reconstruction was done for 2193 genomes, including 469 genomes from this study and samples previously sequenced from Indian laboratories. The genome Wuhan/WH01 (EPI_ISL_406798) was used as the reference for constructing the tree. Clades were assigned according to the nomenclature defined by Nextstrain [15] The resulting phylogenetic tree suggests that out of 469 COVIDSeq genomes, 451 genomes (96%) fell into the A2a clade while 14 genomes (3%) mapped to the I/A3i clade, a distinct clade previously defined for genomes from India [16]. A total of 4 genomes mapped to the B4 clade. The phylogenetic clusters for the genomes are summarised in Fig 5. The distribution of lineages assigned by PANGOLIN suggests a dominant occurrence of the lineages B.1 (n = 286) and B.1.113 (n = 134) as compared to other Indian genomes which show a dominance of B.6 and B.1 lineages. We also found 2 lineages in our dataset, B.1.112 (n = 8) and B.1.99 (n = 1), which have not been previously reported from India. Fig 6 summarises the phylogenetic distribution of the lineages.

The sensitivity of the assay was benchmarked across 649 RT-PCR as well as COVIDSeq confirmed dataset. The analysis revealed the assay had a sensitivity of 97.53% compared to RT-PCR. Since we had only 19 RT-PCR negative samples, we did not assess the specificity of the assay. The comparison of RT-PCR with COVIDSeq assay has been summarized in S5 Table. Notwithstanding, the DRAGEN COVIDSeq protocol identified SARS-CoV-2 in the samples which were negative for RT-PCR for SARS-CoV-2. Additionally, SARS-CoV-2 was detected by the protocol in 21 samples that were inconclusive and 16 samples that were

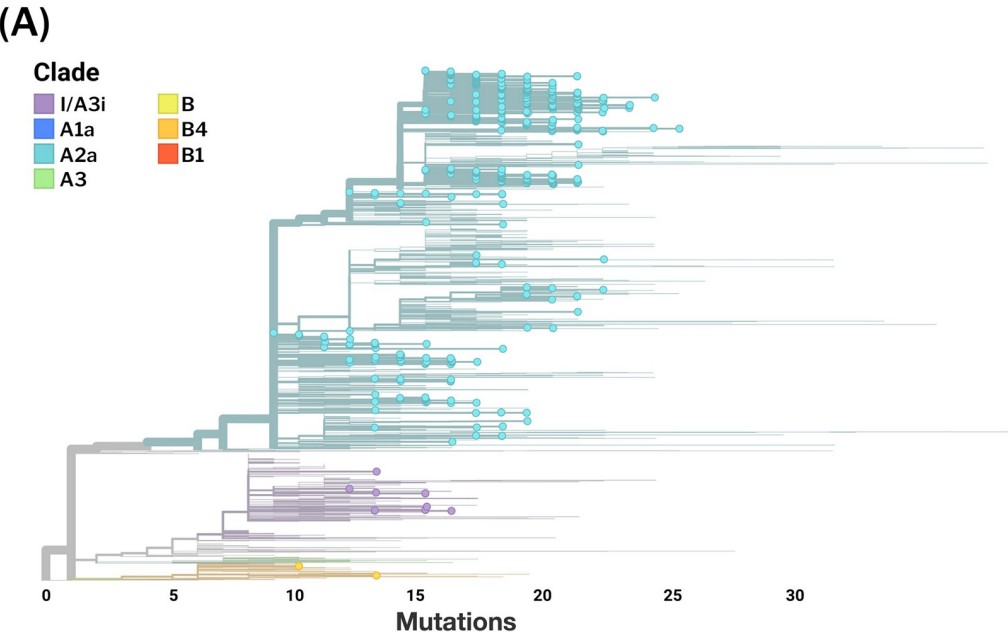

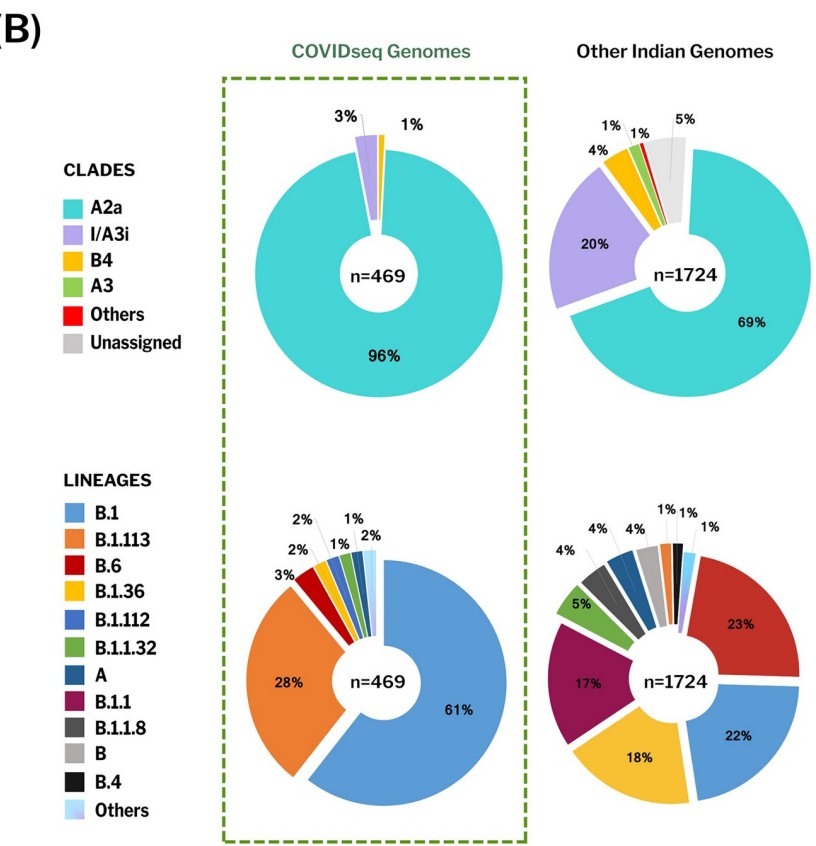

**Fig 5. Phylogenetic distribution of Indian SARS-CoV-2 genomes.** A) Phylogenetic trees generated by Nextstrain. 469 COVIDseq genomes reported from this study are highlighted. The 469 genomes cluster under clade A2a, I/A3i and B4, with A2a being the dominant clade. B) The proportion of clades and PANGOLIN lineages representing the Indian genomes. B.1 and B.1.113 are the dominant lineages in COVIDseq genomes whereas other Indian genomes show a dominance of B.6 and B.1 lineages.

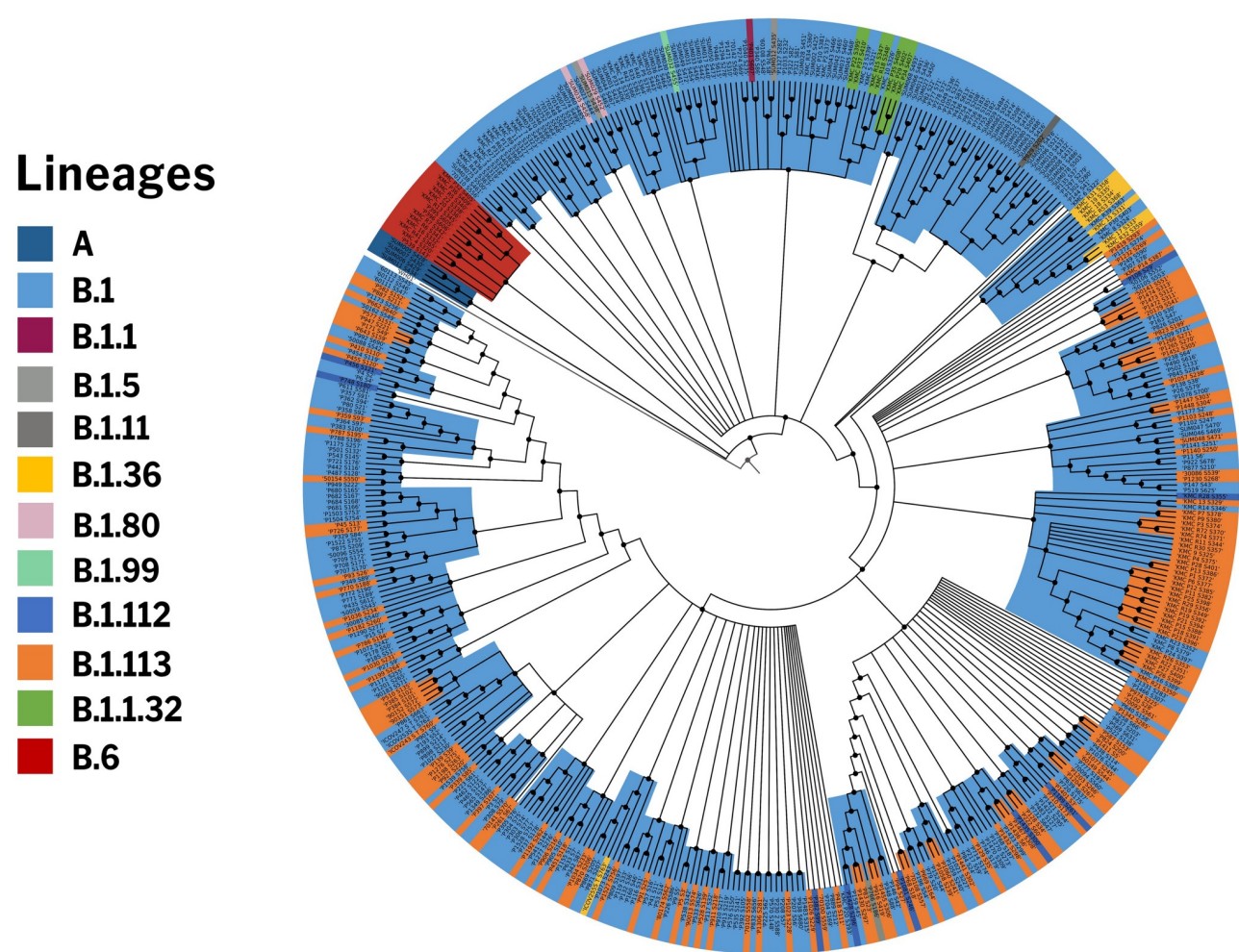

**Fig 6. Phylogenetic distribution of PANGOLIN lineages in COVIDseq genomes.** The distribution of lineages assigned by PANGOLIN in 469 COVIDseq genomes with the Wuhan/WH01 (EPI_ISL_406798) as reference.

annotated pan-sarbeco. We further analysed these samples in great detail to check whether multiple genomic regions were covered in the sequencing experiments. Fig 7 summarises the coverage plots across the genome for the 6 samples which were negative in RT-PCR and detected by COVIDSeq pipeline. The coverage plots for the samples which were inconclusive and pan-sarbeco in RT-PCR were detected by COVIDSeq assay represented in S1 Fig.

Consistently, the samples have over 5% of the genomic region covered in the COVIDSeq protocol suggesting that the protocol could provide for a potentially more sensitive detection assay compared to RT-PCR.

Since the RNA samples were derived from multiple protocols for RNA extraction, we could also get an insight into the compatibility of the protocols with the COVIDSeq test. Of the samples which were SARS-CoV-2 positive on RT-PCR, 182 samples were processed using QIAamp® Viral RNA Mini kit, 264 samples using Nextractor® NX-48S (Genolution, Korea), 201 samples on Trueprep® AUTO v2 (Molbio Diagnostics Pvt. Ltd.), and 8 samples using TRizol based extraction method. Of these, COVIDSeq detected SARS-CoV-2 in 168 of 182 samples (92.3%) from the QIAamp® extracted samples, 263 of 264 samples (99.6%) from the bead-based automated method using Nextractor® NX-48S, 194 of 201 samples (96.5%) from

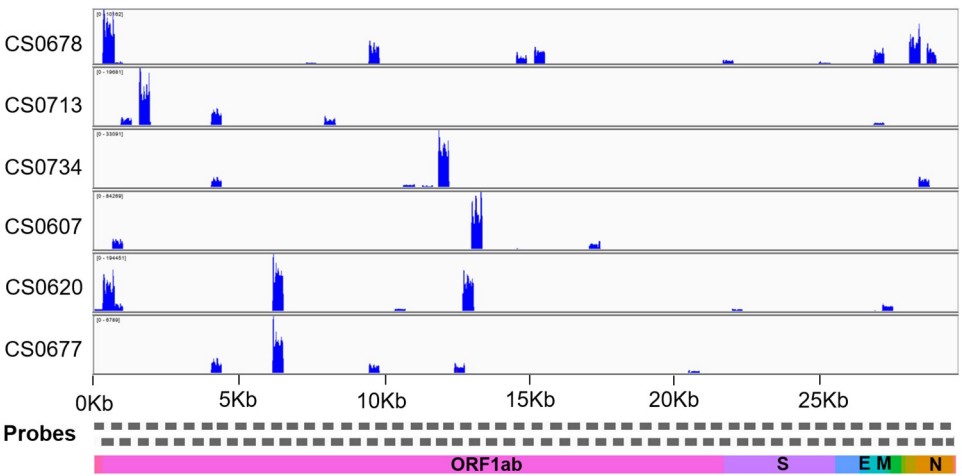

**Fig 7. The coverage plot across the SARS-CoV-2 genome.** The coverage plot constructed using Integrative Genome Viewer (IGV) for samples that were negative on RT-PCR assays but detected by DRAGEN COVIDSeq Pipeline.

## Discussion and conclusions

A number of high-throughput approaches have recently been employed for the detection as well as sequencing of SARS-CoV-2, while RT-PCR based approaches are widely considered as the gold-standard for detection. These include shotgun approaches [17], capture-based [18, 19] as well as amplicon-based [20] approaches followed by next-generation sequencing. Typically the multiplex barcoded library sequencing has been implemented for sample numbers less than 96. A number of approaches have been suggested to increase the throughput of sequencing using barcoded libraries [21, 22]. There is a paucity of data on higher-order multiplex barcoding and sequencing approaches in clinical samples.

In the present report, we evaluated the COVIDSeq approach for high-throughput detection of SARS-CoV-2 which uses multiplex PCR followed by barcoded libraries and sequencing on a next-generation sequencing platform that envisages sequencing 1536 samples per flow cell. We analysed 752 clinical samples in technical duplicates.

Our analysis suggests a high concordance between technical duplicates and a high concordance of detection of SARS-CoV-2 between the COVIDSeq as well as RT-PCR approaches. Our comparative analysis of SARS-CoV-2 detection with RT-PCR and COVIDSeq test showed that the COVIDseq test has comparable sensitivity, precision and accuracy compared to RT-PCR. COVIDSeq protocol detected SARS-CoV-2 in samples previously categorised as inconclusive (21/35), pan-sarbeco (16/43), and negative (6/19) using RT-PCR assays suggesting a comparable sensitivity of the sequencing-based assay compared to RT-PCR. This corresponded to an additional 43/97 samples and a potential gain of 5.71% of samples of the whole dataset and 44.33% of the samples which were considered inconclusive (N = 97), suggesting that the sequencing approach could be used as a potential orthogonal approach to confirm cases that are doubtful or inconclusive in RT-PCR. Notwithstanding the advantage, 16 samples that were annotated positive in RT-PCR were missed in the COVIDSeq approach. Our analysis

also suggests the protocol is compatible with different approaches for RNA isolation suggesting wider applicability in clinical settings where pooling from different labs becomes inevitable.

The COVIDSeq approach additionally provided insights into the genetic epidemiology and evolution of the SARS-CoV-2 isolates. The phylogenetic analysis could be performed for a significantly large number of genomes which gave insights into the prevalent lineage/clades of the virus [23–25]. This analysis also reports two lineages B.1.112 and B.1.99 for the first time in India. B.1.112 and B.1.99 have been previously reported from the USA and UK respectively, implying their origin and distribution beyond India and their possible introductions in India through travelers from these countries, although more data would be needed to confirm this hypothesis.

Furthermore, a total of 1,143 unique variants were contributed by this analysis to the global repertoire of genetic variants. As expected, a significant number of variants were non-synonymous in nature [26, 27]. The present analysis adds a total of 73 novel variants identified for the first time in genomes. Apart from the throughput of sample analysis, the COVIDSeq approach is also remarkable in terms of speed, with a sequencing time of 11 hours and an analysis timeline of 6 hours. Given that the NovaSeq6000 sequencer used in the present study can handle two S4 flow cells in parallel, this could be potentially scaled to a throughput of 1536x2 samples that can be handled in parallel.

In conclusion, our analysis suggests that COVIDSeq is a high-throughput sequencing-based approach that is sensitive for the detection of SARS-CoV-2. In addition, COVIDSeq has an additional advantage of enabling the genetic epidemiology of SARS-CoV-2.

## Materials and methods

### Patients and samples

The study was approved by the Institutional Human Ethics Committee of CSIR-Institute of Genomics and Integrative Biology (IHEC No. Dated CSIR-IGIB/IHEC/2020-21/01) and the patient consent has been waived by the ethics committee. Samples from nasal, nasopharyngeal, and oropharyngeal swabs were obtained according to the standard protocol and collected in 3 ml sterile viral transport medium (VTM) tube or 1ml of TRIzol reagent (Invitrogen). All the samples were transported to the laboratory at a cold temperature (2–8˚C) within 72 hours post collection, and stored at -80˚C till further used.

### RNA isolation

RNA extraction was carried out in a pre-amplification environment with a Biosafety level 2 (BSL-2) facility. RNA isolation was done using four different methods. For manual RNA extraction, a total of 140 μl of the VTM medium was used; prior to isolation, the VTM samples were subjected to heat inactivation at 50˚C for 30 minutes. After heat inactivation, the RNA was extracted from 140 μl of VTM samples using QIAamp® Viral RNA Mini kit (QIAGEN) as per the manufacturer's instructions. For the automated magnetic bead-based extraction method, 200 μl of VTM was transferred to a 96-well deep well cartridge plate supplied with the kit (VN143), and extraction was performed on Nextractor® NX-48S instrument (Genolution Inc.) as instructed by the manufacturer. After a bead-based capture and washing process, the RNA sample was eluted in 40 μl of the elution buffer. For RNA isolation using Trueprep AUTO v2 universal cartridge-based sample prep device, (Molbio Diagnostics Pvt. Ltd.) 500 μl of the VTM was added to the 2.5 ml of lysis buffer provided with the kit. After pipette mixing, 3 ml of the mixture was dispensed in the provided cartridge; the final RNA was eluted in 50 μl of elution buffer. For RNA from TRIzol reagent, the tubes containing swabs were vortexed briefly. The overall content of the TRIzol tube was transferred into a 1.5 ml tube, followed by

the addition of 200 μl of chloroform and mixed by inverting the tubes several times. After 5 minutes of incubation, the 1.5 ml tubes were centrifuged for 15 minutes at 12,000 RPM at 4˚C. The upper clear aqueous layer which contains the RNA was transferred to new tubes. An equal amount of isopropanol was added to the tubes containing the RNA. Contents of the tubes were mixed by inverting the tubes several times and tubes were incubated for 10 minutes on ice followed by centrifugation for 10 minutes at 12,000 RPM at 4˚C. The supernatant was discarded and the RNA pellet was dissolved in 30 μl of RNase-free water after 2 ethanol washes. TURBO DNase (Ambion, Applied Biosystems) treatment was given to the isolated RNA to remove genomic DNA contamination in the samples followed by RNA purification using the phenol/chloroform method.

## Real-time PCR for SARS-CoV-2

To detect SARS-CoV-2 viral infection, a one-step Real-Time PCR assay was performed using STANDARD M nCoV Real-Time detection kit (SD Biosensor, Korea), targeting the nCoV2 specific ORF1ab (RdRp) and pan-sarbeco specific E genes on LightCycler® 480 System (Roche) and ABI 7500 Fast DX (Applied Biosystems) as per the manufacturer's instructions.

## Library preparation and sequencing

The libraries were prepared using Illumina COVIDSeq protocol (Illumina Inc, USA). The first strand synthesis was carried out on RNA samples isolated using different extraction methods individually, in Biosafety level 2 (BSL-2) plus environment following standard protocols. The synthesized cDNA was amplified using a multiplex polymerase chain reaction (PCR) protocol, producing 98 amplicons across the SARS-CoV-2 genome (https://artic.network/). The primer pool additionally had primers targeting human RNA, producing an additional 11 amplicons. The PCR amplified product was later processed for tagmentation and adapter ligation using IDT for Illumina Nextera UD Indexes Set A, B, C, D (384 indexes, 384 samples). Further enrichment and cleanup was performed as per protocols provided by the manufacturer (Illumina Inc). All samples were processed as batches in a 96-well plate that consisted of one of COVIDSeq positive control HT (CPC HT) and one no template control (NTC); these 96 libraries were pooled together in a tube. Pooled samples were quantified using Qubit 2.0 fluorometer (Invitrogen Inc.) and fragment sizes were analyzed in Agilent Fragment analyzer 5200 (Agilent Inc). The pooled library was further normalized to 4nM concentration and 25 μl of each normalized pool containing index adapter set A, B, C, and D were combined in a new microcentrifuge tube to a final concentration of 100pM and 120pM. For sequencing, pooled libraries were denatured and neutralized with 0.2N NaOH and 400mM Tris-HCL (pH-8). Replicas of each 384 sample pools were loaded onto the S4-flow cell following NovaSeq-XP workflow as per the manufacturer's instructions (Illumina Inc). Dual indexed single-end sequencing with 36bp read length was carried out on NovaSeq 6000 platform.

## Data processing

The raw data generated in binary base call (BCL) format from NovaSeq 6000 was processed using DRAGEN COVIDSeq Test Pipeline (Illumina Inc.) on the Illumina DRAGEN v3.6 Bio-IT platform as per standard protocol. The analysis involves sample sheet validation, data quality check, FASTQ generation, and SARS-CoV-2 detection when at least 5 SARS-CoV-2 probes are detected. Further samples with SARS-CoV-2 and at least 90 targets detected were processed for alignment, variant calling and consensus sequence generation.

For an in-depth analysis, we additionally analysed the data using a custom pipeline. This included demultiplexing the raw data to FASTQ files using bcl2fastq (v2.20) followed by a

quality assessment of the FASTQ files using Trimmomatic (v0.39) [28]. An average base quality of Q30 and read length cut-off of 30 bps were used for trimming, apart from the adapter sequences. We followed a recently published protocol to perform reference-based assembly [29]. As per protocol, the trimmed reads were aligned to the human reference genome (GRCh38 / hg38) and severe acute respiratory syndrome coronavirus 2 (SARS-CoV-2) Wuhan-Hu-1 reference genome (NC_045512.2) using HISAT2-2.1 [30]. The reads mapped to hg38 were further discarded and the unaligned reads were extracted using samtools (v 1.10) [31]. The unaligned reads were further mapped to the Wuhan Hu-1 genome and the alignment statistics were evaluated [3]. The data was merged for duplicates for the variant calling and consensus sequence generation. Variant calling was performed using VarScan (v2.4.4) for samples with genome coverage greater than 99% [32]. Samtools (v 1.10) [31], bcftools (v 1.10.2), and seqtk (version 1.3-r114) [33] were used to generate the consensus sequence. We have also evaluated the correlation coefficient with a p-value <0.01 between the duplicates total reads and genome coverage.

## Annotation of genetic variants and comparison with existing datasets

The variants were systematically annotated using ANNOVAR [34]. Annotations on genomic loci and functional consequences of the protein were retrieved from RefSeq. Custom databases were created for annotations on functional consequences, potential immune epitopes, protein domains, and evolutionary conservation scores. Genomic loci associated with common error-prone sites and diagnostic primer/probe regions were manually curated and were systematically converted to data tables compatible with ANNOVAR for added annotation options. The functional relevance of the identified genetic variants was also analyzed. The variants were systematically compared with a manually curated compendium of SARS-CoV-2 variations with potential functional impacts compiled from literature reports [35]. All the filtered variants were checked with other viral genomes submitted from India and worldwide. Genomes with alignment percentage of at least 99 and gap percentage <1 were filtered as high quality. A total of 1372 high-quality genomes from India out of 1888 submitted till July 28, 2020, were included in the analysis. Similarly, global genomes submitted till August 07, 2020, were included, accounting for 29,177 high-quality genomes out of 79,764. Details of the samples, originating and submitting laboratories are listed in the S7 Table. Mutation information provided by Nextstrain [33] till August 08, 2020, was also used for comparison.

## Phylogenetic analysis

A total of 495 samples that had at least 99% genome coverage were considered for this analysis, along with the dataset of SARS-CoV-2 genomes from India deposited in GISAID. The sample names and the name of the originating and submitting institutions are listed in S8 Table. We followed a previously described protocol for phylogenetic clustering [36]. A total of 26 COVIDSeq genomes having Ns > 5% were removed from the analysis. Genomes from GISAID having Ns > 5% and ambiguous dates of sample collection were also excluded from the analysis. The phylogenetic network was built using the analysis protocol for SARS-CoV-2 genomes provided by Nextstrain [15]. The genome sequences were aligned using MAFFT to the reference genome and problematic variant positions were masked [37]. A raw phylogenetic tree was constructed using IQTREE and the raw tree was refined to construct a molecular-clock phylogeny, infer mutations, and identify clades [38]. The resulting phylogenetic tree was viewed using Auspice, an interactive visualization web-application provided by Nextstrain. Lineages were also assigned to the genomes using the Phylogenetic Assignment of Named

Global Outbreak LINeages (PANGOLIN) package [39]. The phylogenetic distribution of the lineages was visualized and annotated using iToL [40].

## Comparison of RT-PCR test with the sequencing-based COVIDSeq test

Initially, all the samples underwent RT-PCR based screening for the presence of SARS-CoV-2 RNA. Out of these 752 samples, 655 (87.1%) samples were RT-PCR positive, 43 (5.7%) were pan-sarbeco, 35 (4.6%) were inconclusive and 19 (2.5%) were negative. We compared the sample type (e.g. positive, pan-sarbeco, inconclusive, and negative) WGS output and calculated the percent of the genome covered, sensitivity, specificity, accuracy, precision, and gain of detection rate. The methodology adopted in this study has been represented in Fig 1.

## Supporting information

**S1 Fig. Coverage plots for 37 samples that were inconclusive and pan-sarbeco by RT-PCR and detected positive for SARS-CoV-2 by sequencing.**
(TIF)

**S1 Table. DRAGEN COVIDSeq test pipeline time summary for each task.**
(PDF)

**S2 Table. Data summary of the COVIDSeq, RT-PCR, and custom pipeline analysis.** NA-Not Applicable.
(PDF)

**S3 Table. Summary of functional annotation of the genetic variants reported in the study.**
(PDF)

**S4 Table. Compilation of details of genetic variants with reported functional relevance.**
(PDF)

**S5 Table. Summary of the COVIDSeq assay comparison with RT-PCR.**
(PDF)

**S6 Table. Comparison of different RNA extraction methods and detection of the SARS-CoV-2 with RT-PCR and COVIDSeq test.**
(PDF)

**S7 Table. GISAID acknowledgement table for global genomes used in the study.**
(PDF)

**S8 Table. GISAID acknowledgement table for Indian genomes used in the study.**
(PDF)

## Acknowledgments

Authors acknowledge Samatha Mathew for reviewing the manuscript and providing constructive criticism. Authors acknowledge Vigneshwar Senthivel for creating the graphical abstract of the COVIDSeq pipeline. AJ, MD, BJ, PS acknowledges research fellowships from CSIR and MP acknowledges a research fellowship from BINC-DBT. DS acknowledges a research fellowship from Intel. The authors SS, MS, AN acknowledge the SOAU for their PhD fellowship.

## Author Contributions

**Conceptualization:** Rahul C. Bhoyar, Vinod Scaria, Sridhar Sivasubbu.

**Data curation:** Rahul C. Bhoyar, Abhinav Jain, Paras Sehgal, Disha Sharma, Mercy Rophina, Swayamprabha Sahoo, Maheswata Sahoo, Rahul Sahlot, Avileli Surekha, Vinod Scaria, Sridhar Sivasubbu.

**Formal analysis:** Rahul C. Bhoyar, Abhinav Jain, Mohit Kumar Divakar, Disha Sharma, Mohamed Imran, Vinod Scaria.

**Funding acquisition:** Vinod Scaria, Sridhar Sivasubbu.

**Investigation:** Rahul C. Bhoyar, Vinod Scaria, Sridhar Sivasubbu.

**Methodology:** Rahul C. Bhoyar, Mohit Kumar Divakar, Mohamed Imran, Gyan Ranjan, Sumit Sharma, Sanjay Siwach, Ananya Nayak, Jatindra Nath Mohanty, Jayashankar Das, Sudhir Bhandari, Sandeep K. Mathur, Anshul Kumar, Pallavali Rojarani, Juturu Vijaya Lakshmi, Pulala Chandra Sekhar, Shelly Mahajan, Shet Masih, Pawan Singh, Vipin Kumar, Blessy Jose, Vidur Mahajan, Vivek Gupta, Rakesh Gupta, Prabhakar Arumugam, Anjali Singh, Ananya Nandy, Ragavendran P. V., Rakesh Mohan Jha, Anupama Kumari, Sheetal Gandotra, Vivek Rao, Mohammed Faruq, Sanjeev Kumar, Betsy Reshma G., Narendra Varma G., Shuvra Shekhar Roy, Antara Sengupta, Sabyasachi Chattopadhyay, Khushboo Singhal, Shalini Pradhan, Diksha Jha, Salwa Naushin, Saruchi Wadhwa, Nishu Tyagi, Mukta Poojary, Vinod Scaria, Sridhar Sivasubbu.

**Project administration:** Rahul C. Bhoyar, Vinod Scaria, Sridhar Sivasubbu.

**Resources:** Vinod Scaria, Sridhar Sivasubbu.

**Software:** Abhinav Jain, Bani Jolly, Mercy Rophina, Kavita Pandhare.

**Supervision:** Rahul C. Bhoyar, Vinod Scaria, Sridhar Sivasubbu.

**Validation:** Rahul C. Bhoyar.

**Visualization:** Bani Jolly, Vinod Scaria.

**Writing – original draft:** Rahul C. Bhoyar, Abhinav Jain, Vinod Scaria, Sridhar Sivasubbu.

**Writing – review & editing:** Rahul C. Bhoyar, Abhinav Jain, Paras Sehgal, Vinod Scaria, Sridhar Sivasubbu.

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
