## [Decision Letter · Decision Letter 0]

20 Oct 2020

PONE-D-20-25162

High throughput detection and genetic epidemiology of SARS-CoV-2 using COVIDSeq next generation sequencing

PLOS ONE

Dear Dr. Scaria,

Thank you for submitting your manuscript to PLOS ONE. After careful consideration, we feel that it has merit but does not fully meet PLOS ONE’s publication criteria as it currently stands. Therefore, we invite you to submit a revised version of the manuscript that addresses the points raised during the review process.

Please find a summary of the reviews appended below. The reviewers recommend reconsideration of your paper following major revision. I invite you to resubmit your manuscript after addressing all reviewer comments. When resubmitting your manuscript, please carefully consider all issues mentioned in the reviewers' comments, outline every change made point by point, and provide suitable rebuttals for any comments not addressed.

We look forward to receiving your revised manuscript.

Kind regards,

Obul Reddy Bandapalli, MSc, PhD

Academic Editor

PLOS ONE

Journal Requirements:

'The study was approved by the Institutional Human Ethics Committee (IHEC No. Dated CSIR-IGIB/IHEC/2020-21/01).'

3. Please provide additional details regarding participant consent.

In the ethics statement in the Methods and online submission information, please ensure that you have specified (a) whether consent was informed and (b) what type you obtained (for instance, written or verbal, and if verbal, how it was documented and witnessed).

If the need for consent was waived by the ethics committee, please include this information.

Reviewers' comments:

Reviewer's Responses to Questions

**Comments to the Author**

1. Is the manuscript technically sound, and do the data support the conclusions?

Reviewer #1: Yes

Reviewer #2: Yes

2. Has the statistical analysis been performed appropriately and rigorously? 

Reviewer #1: Yes

Reviewer #2: Yes

3. Have the authors made all data underlying the findings in their manuscript fully available?

Reviewer #1: Yes

Reviewer #2: Yes

4. Is the manuscript presented in an intelligible fashion and written in standard English?

Reviewer #1: Yes

Reviewer #2: Yes

5. Review Comments to the Author

Reviewer #1: Authors has done reasonable and robust approaches for identification of SARSCOV2 using NSG platforms and validated them in few patients.

However i have following concerns which need attention by the reviewers and I believe addressing this issue would enhance the validity of the work

1. How we can verify the analysis done at genetic levels with proteins based information.

2. Whether approach actually translate into proteins levels and whether authors identified some of the major antigens which actually bind to the ACE2 receptors.

3. I would advice author to pickup clinical isolate of the covid 19 , infect VeroE6 cells and do the analysis of these major determinant of the virus (ORF) which are responsible not only for the diagnosis but also for the viable infection in host

Minor revision

Reviewer #2: The manuscript submitted by Bhoyar et al. have used COVIDseq NGS to detect SARS-CoV-2, found new variants and performed a phylogenetic variation. It is a good study with information on evolution of SARS-CoV-2, which can be used in the subsequent studies to understand the impact of specific mutations. Given the ongoing pandemic, I consider that the study is of high significance. However, I have listed some comments for the authors to address.

1. The authors claim that this method can be used as a confirmatory test and a potential high sensitivity assay for detection of SARS-CoV-2. Also, they mention in their discussion that “COVIDseq test ourpeformed with increased sensitivity, precision and accuracy compared to RT-PCR”. I agree that it can be used as a large-scale rapid detection method/confirmatory test but I have a concern regarding their claim with respect to its sensitivity/precision/accuracy compared to RT-PCR, based on the data given in Supplementary Table 4a. I see that 16/655 RT-PCR positive positive samples were not detected in comparison to detecting the viral RNA in 6/655 RT-PCR negative samples. Similarly, 25/43 Pan-Sarbeco positive samples were not detected by their method. It would be helpful if authors can address this concern and discuss/state accordingly in their manuscript. Will it be fair to say that this method has comparable sensitivity with RT-PCR?

Also, it would be helpful if authors can discuss the possible technical reasons for discrepancy between the tests.

2. I am wondering if the authors can comment on the geographical distribution/chronology of the samples they analyzed and the genetic variants of the SATS-CoV-2, they detected. Also, it might be helpful if they can elaborate in their discussion on phylogenetic clades them mention about. They are named so first time or they are using clades name based on previous studies? It is not clear. It will also be helpful if they can include in their discussion what the new clades/mutation rates mean for the pandemic in India.

3. For the benefit of readers from diverse background a brief description on pan-sarbeco testing should be included.

4. The authors mention that 4 different RNA extraction methods were used for preparing cDNA. It is not clear if they combined RNA from these methods or carried out testing separately. Clarification would be helpful.

5. A suggestion for future consideration: I am wondering if authors are losing some RNA during phenol-chloroform purification after DNAse treatment. Would it worth exploring TURBO kit to remove the DNAse, which does not involve additional purification step like phenol-choloroform method.

Minor editing needed:

- Line 74-75 “this approach will become more effective….”. May be it is a good idea to say “is expected to become more effective…”

- In several places hyphen is missing. E.g. line 79 it should be nucleic acid-based and sequencing-based. Address the same throughout the manuscript where hyphen is missing.

- Where non-standard abbreviations appear first in the manuscript, should be expanded. E.g. line 119 CPC HT. Please address the same throughout the manuscript.

6. PLOS authors have the option to publish the peer review history of their article (what does this mean?). If published, this will include your full peer review and any attached files.

Reviewer #1: No

Reviewer #2: No

---

## [Author Response · Author response to Decision Letter 0]

15 Jan 2021

RESPONSE TO REVIEWERS’ COMMENTS

5. Review Comments to the Author

Reviewer #1: Authors have done reasonable and robust approaches for identification of SARSCOV2 using NSG platforms and validated them in a few patients.

However i have following concerns which need attention by the reviewers and I believe addressing this issue would enhance the validity of the work

1. How we can verify the analysis done at genetic levels with proteins based information.

ANS: For all the genetic variants identified in the study, and are non-synonymous in nature, we have created a comprehensive annotation for the amino acid changes, computationally predicted functional annotations on protein domains, immune epitopes as well as their frequencies. This annotation table is available as S3 Table.

2. Whether the approach actually translates into protein levels and whether authors identified some of the major antigens which actually bind to the ACE2 receptors.

ANS: Our analysis has indeed revealed a number of genetic variants in the receptor binding domain of S protein. Some of the genetic variants with functional annotations have also been retrieved from FaviCov (Rophina et al 2020) and are summarised. This annotation table is available as S4 Table.

3. I would advice author to pick up clinical isolate of the covid 19 , infect VeroE6 cells and do the analysis of these major determinant of the virus (ORF) which are responsible not only for the diagnosis but also for the viable infection in host

ANS: We apologize this experiment would be beyond the scope of the manuscript, which intends to describe an approach for detection of SARS-CoV-2 rather than understanding the mechanisms of infection.

Minor revision

Reviewer #2: The manuscript submitted by Bhoyar et al. have used COVIDseq NGS to detect SARS-CoV-2, found new variants and performed a phylogenetic variation. It is a good study with information on evolution of SARS-CoV-2, which can be used in the subsequent studies to understand the impact of specific mutations. Given the ongoing pandemic, I consider that the study is of high significance. However, I have listed some comments for the authors to address.

1. The authors claim that this method can be used as a confirmatory test and a potential high sensitivity assay for detection of SARS-CoV-2. Also, they mention in their discussion that “COVIDseq test outperformed with increased sensitivity, precision and accuracy compared to RT-PCR”. I agree that it can be used as a large-scale rapid detection method/confirmatory test but I have a concern regarding their claim with respect to its sensitivity/precision/accuracy compared to RT-PCR, based on the data given in Supplementary Table 4a. I see that 16/655 RT-PCR positive positive samples were not detected in comparison to detecting the viral RNA in 6/655 RT-PCR negative samples. Similarly, 25/43 Pan-Sarbeco positive samples were not detected by their method. It would be helpful if authors can address this concern and discuss/state accordingly in their manuscript. Will it be fair to say that this method has comparable sensitivity with RT-PCR?

ANS: We thank the reviewer for the comments. We have now changed the statement to reflect the observations. The text now reads. The COVIDSeq test has comparable sensitivity, precision and accuracy compared to RT-PCR.

Also, it would be helpful if authors can discuss the possible technical reasons for discrepancy between the tests.

ANS: We have now retrieved 13 samples which were positive on RT-PCR and negative on COVID-seq for which we had RNA available and performed RT_PCR. Our analysis revealed that a significant number (N=10) had RNA degraded. We surmise therefore that RNA degradation could be an issue with the false negatives on COVIDseq.

2. I am wondering if the authors can comment on the geographical distribution/chronology of the samples they analyzed and the genetic variants of the SARS-CoV-2, they detected. Also, it might be helpful if they can elaborate in their discussion on phylogenetic clades them mention. They are named so first time or they are using clades name based on previous studies? It is not clear. It will also be helpful if they can include in their discussion what the new clades/mutation rates mean for the pandemic in India.

ANS: We thank the reviewer for pointing this out. We have now added a supplementary data summarising the geographical origin of the samples and the time of collection. The clade names have been following the NextStrain as well as PANGOLIN clade classifications which is widely followed in literature.

3. For the benefit of readers from diverse backgrounds a brief description on pan-sarbeco testing should be included.

ANS; We thank the reviewer for the suggestion. We have now detailed pan-sarbeco in the materials and methods sections as well as the results section.

4. The authors mention that 4 different RNA extraction methods were used for preparing cDNA. It is not clear if they combined RNA from these methods or carried out testing separately. Clarification would be helpful.

ANS; We thank the reviewer for pointing out this, and we agree the statement could be confusing. We have now revised the text to explain the methodology in detail in the Materials section.

5. A suggestion for future consideration: I am wondering if authors are losing some RNA during phenol-chloroform purification after DNAse treatment. Would it be worth exploring the TURBO kit to remove the DNAse, which does not involve additional purification steps like the phenol-chloroform method.

ANS: We thank the reviewer for the important suggestion. As per the comment we have taken utmost precautions during the purification step towards minimizing the RNA loss using phenol-chloroform method. Further, we have checked the concentration of the purified RNA using nanodrop instrument, and concentrations of all the isolated RNA were falling in the 20-50ng/µl range. Apart from that we have also run the RT-PCR on the same RNA samples, which gives the consistent result. However, we agree with the reviewers suggestion and we will definitely adopt the suggested approach for RNA purification in our future experiments. But we experienced a shortcoming of the turbo DNAse kit, that minute impurities of beads used in the assay affect the single step RT reaction. We are trying to modify the protocol to resolve the issue and to adapt it in future experiments.

Minor editing needed:

- Line 74-75 “this approach will become more effective….”. Maybe it is a good idea to say “is expected to become more effective…”

ANS; Thanks for pointing this out. We have now made the corrections as suggested in the revised manuscript.

- In several places hyphens are missing. E.g. line 79 it should be nucleic acid-based and sequencing-based. Address the same throughout the manuscript where hyphen is missing.

ANS; Thanks for pointing this out. We have now made the corrections as suggested in the revised manuscript.

- Where non-standard abbreviations appear first in the manuscript, should be expanded. E.g. line 119 CPC HT. Please address the same throughout the manuscript.

ANS; Thanks for pointing this out. We have now made the corrections as suggested in the revised manuscript.

---

## [Decision Letter · Decision Letter 1]

2 Feb 2021

High throughput detection and genetic epidemiology of SARS-CoV-2 using COVIDSeq next-generation sequencing

PONE-D-20-25162R1

Dear Dr. Scaria,

We’re pleased to inform you that your manuscript has been judged scientifically suitable for publication and will be formally accepted for publication once it meets all outstanding technical requirements.

Kind regards,

Obul Reddy Bandapalli, MSc, PhD

Academic Editor

PLOS ONE

Additional Editor Comments (optional):

Reviewers' comments:

Reviewer's Responses to Questions

**Comments to the Author**

1. If the authors have adequately addressed your comments raised in a previous round of review and you feel that this manuscript is now acceptable for publication, you may indicate that here to bypass the “Comments to the Author” section, enter your conflict of interest statement in the “Confidential to Editor” section, and submit your "Accept" recommendation.

Reviewer #1: All comments have been addressed

Reviewer #2: All comments have been addressed

2. Is the manuscript technically sound, and do the data support the conclusions?

Reviewer #1: Yes

Reviewer #2: Yes

3. Has the statistical analysis been performed appropriately and rigorously? 

Reviewer #1: Yes

Reviewer #2: Yes

4. Have the authors made all data underlying the findings in their manuscript fully available?

Reviewer #1: Yes

Reviewer #2: Yes

5. Is the manuscript presented in an intelligible fashion and written in standard English?

Reviewer #1: Yes

Reviewer #2: Yes

6. Review Comments to the Author

Reviewer #1: Authors has addressed all of my concerns , the manuscript may be acceptable for the publication after grammatical checks

Reviewer #2: (No Response)

7. PLOS authors have the option to publish the peer review history of their article (what does this mean?). If published, this will include your full peer review and any attached files.

Reviewer #1: No

Reviewer #2: No

---

## [Editor Report · Acceptance letter]

5 Feb 2021

PONE-D-20-25162R1 

High throughput detection and genetic epidemiology of SARS-CoV-2 using COVIDSeq next-generation sequencing 

Dear Dr. Scaria:

I'm pleased to inform you that your manuscript has been deemed suitable for publication in PLOS ONE. Congratulations! Your manuscript is now with our production department. 

Kind regards, 

on behalf of

Dr. Obul Reddy Bandapalli 

Academic Editor

PLOS ONE